materials science/energy

plasmonics, solar cell, silicon

**Author for correspondence:**
Mohammed Shahriar Sabuktagin
e-mail: s1hahriar@yahoo.com

# Large plasmonic absorption enhancement effect of triangular silver nanowires in silicon

Mohammed Shahriar Sabuktagin[1]
and Khairus Syifa Hamdan[2]

[1]Department Electrical and Electronic Engineering, Uttara University, Dhaka 1230, Bangladesh
[2]UM Power Energy Dedicated Advanced Centre (UMPEDAC), University of Malaya, 50603 Kuala Lumpur, Malaysia

(iD) MSS, 0000-0003-2240-9811

Two-dimensional finite difference time domain (FDTD) simulations were performed for evaluating optical absorption enhancement and loss effects of triangular silver (Ag) nanowires embedded in silicon (Si) thin-film photovoltaic device structures. Near-bandgap absorption enhancement in Si was much larger than the reported values of other nanostructures from similar simulations. A nanowire with equal sides of 20 nm length showed 368-fold absorption enhancement whereas only 5× and 15× enhancement were reported for solid spherical and two-dimensional core-shell type nanostructures, respectively. Undesirable absorption loss in the metal of the nanowire was 3.55× larger than the absorption in Si which was comparable to the value reported for the spherical nanoparticle. Interestingly, as the height of the nanowire was increased to form a sharper tip, absorption loss showed a significant drop. For a nanowire with 20 nm base and 20 nm height, absorption loss was merely 1.91× larger than the absorption in Si at the 840 nm plasmon resonance. This drop could be attributed to weaker plasmon resonance manifested by lower metallic absorption in the spatial absorption map of the nanowire. However, absorption enhancement in Si was still large due to strong plasmonic fields at the sharper and longer tip, which was effective in enhancing absorption over a larger area in Si. Our work shows that the shape of a nanostructure and its optimization can significantly affect plasmonic absorption enhancement and loss performance in photovoltaic applications.

## 1. Introduction

Thin-film photovoltaic devices are typically too thin to absorb solar radiation completely at longer wavelengths. Consequently, a good

This article has been edited by the Royal Society of Chemistry, including the commissioning, peer review process and editorial aspects up to the point of acceptance.

portion of solar energy escapes through the device without being converted into electricity [1]. Still, thin-film devices are economically and environmentally attractive due to lower material requirements and manufacturing costs. Hence there is strong interest in improving performance of these devices, which is being pursued in a number of ways including device design, material quality improvement and enhancement of sunlight absorption [2,3]. Being the second most earth abundant material, Si is a prime candidate for large-scale photovoltaic energy generation. A number of light trapping techniques have been proposed for increasing absorption [3–5] in thin-film solar cell structures. Plasmonic light trapping method makes use of incident sunlight-induced plasmon resonance in metal nanostructures for enhancing absorption. Absorption may be increased by locally enhanced electric field intensities near the surface of the nanostructure and light scattering by the nanostructure into the solar cell structure. A variety of metal nanostructures have been explored for plasmonic enhancement. Finite difference time domain (FDTD) simulations showed up to 5× and 15× enhancements for three-dimensional solid spherical nanoparticles [6] and two-dimensional core-shell type nanowires [7], respectively. Analytical calculations based on Mie theory indicated the possibility of nearly 400× enhancement [8]. Dramatic enhancement of electric field intensities were reported for triangular nanowires in [9]. However, plasmonic enhancement is not free from drawbacks. It has been shown that undesirable absorption loss in the metal of a plasmonic nanostructure can be much larger than the desirable absorption enhancement in the semiconductor [6]. This is because electric field intensity gets enhanced in the metal of the nanostructure too and optical absorption depends on electric field intensity as well as the imaginary component of permittivity of the material. Since this component in metals is quite large at optical frequencies, undesirable absorption loss in the metal can be larger than the enhanced absorption in the semiconductor. Such larger loss of spherical Ag nanoparticles embedded in Si prompted the authors in [6] to express pessimistic opinion regarding the prospect of plasmonic absorption enhancement for Si photovoltaic devices. Hence, it is highly desirable to find ways of reducing this loss. Various loss reduction strategies have been suggested in the literature. Authors in [7] suggested that core-shell type nanostructures may show lower loss due to reduction of metal content by the dielectric core. Development of low loss plasmonic materials was proposed in [10]. On the other hand, it has been shown that variation of nanostructure [11,12] and tip shapes [13,14] can cause large variations in plasmonic properties. In this work, we show that the shape details can significantly affect the plasmonic absorption enhancement and loss performance as well. These results were obtained from two-dimensional FDTD simulations of triangular Ag nanowires embedded in Si thin-film structures. Simulations showed that enhancement and loss performance of triangular nanowires were much better than those reported for spherical [6] and core-shell type nanostructures [7]. Additionally, a large reduction of absorption loss was observed as the shape of the triangle was varied to form a sharper tip. We speculate that these trends may be present in three-dimensional nanostructures with sharp tips as well, which could show even larger enhancement and lower loss. If the material loss could be lowered to some extent following the approaches proposed in [10], use of those low-loss materials for fabricating the nanowires discussed in this study could possibly lead to much lower loss than absorption even in the low absorption band in Si. Plasmonic enhancement of impurity photovoltaic effects [15,16] may also benefit from the large enhancement capability of these triangular nanowires. Our work shows that selection of shape and its tuning can be important factors in achieving large absorption enhancement at the cost of small loss.

## 2. Simulation methodology

Simulations were performed in two dimensions using Lumerical FDTD Solutions (https://www.lumerical.com/). Ag and Si material models were based on data from Palik [17]. Staircase approximation was applied at material boundaries. The metallic regions of the nanowires was meshed at 0.05 nm resolution.

Figure 1 shows the simulation set-up. A single Ag nanowire was placed at the centre of a square-shaped simulation region with sides of 1 $\mu$m length. This region was filled with Si and truncated by perfectly matched layer (PML) boundaries. Localized plasmon modes in the nanowire was excited using a total field scattered field (TFSF) source with transverse magnetic (TM) polarization [18]. Similar simulation set-up was used for analysing core-shell type nanowires in [7]. Power absorption was recorded in each Yee cell of a 40 by 40 nm box-shaped region with the nanowire at the centre. This was done using a feature in Lumerical software called 'Advanced Power Absorption'. Power absorption was calculated from the electric field values obtained from the FDTD simulations and the imaginary component of permittivity of a Yee cell according to the formula

$$P_{abs} = -0.5 \; \omega |E|^2 \; \text{imag}(\varepsilon).$$

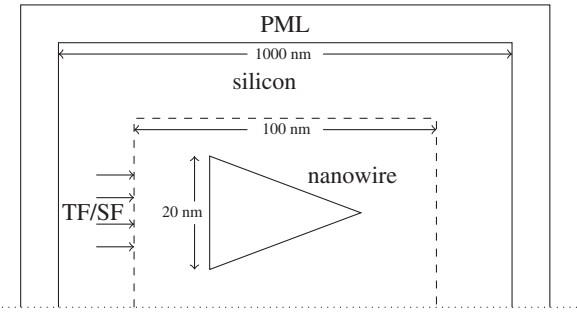

**Figure 1.** Partial view of the simulation set-up.

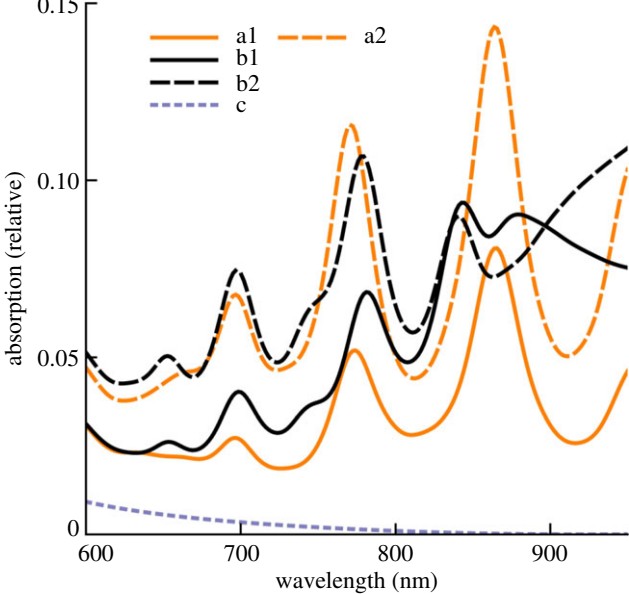

**Figure 2.** Normalized (with respect to source power) absorption versus wavelength plots. (a1) Absorption in Si due to the equilateral nanowire. (a2) Absorption loss in the metal of the equilateral nanowire. (b1) Absorption in Si due to the sharp-tipped bilateral nanowire. (b2) Absorption loss in the metal of the bilateral nanowire. (c) Absorption in bare Si.

Total absorption in Ag and Si at a particular wavelength was determined by summing absorption in the Yee cells located within the respective materials. This was accomplished in a post-processing step using scripts available from Lumerical. Absorption at every wavelength was normalized with respect to source power.

Incorporation of these nanowires in large-area solar cell devices may be possible using nanoimprint lithography techniques [19–22]. We suggest development of substrates with multiple layers of nanowires for complete absorption of solar radiation in the solar cell. The nanowires may be fabricated on a glass substrate with alternating square-shaped patterns of two polymers which are soluble in different solvents [23]. Thickness and lateral dimensions of the polymer patterns can be 100 nm and 1 μm, respectively, for example. This may be repeated multiple times to obtain a substrate with multiple layers of nanowires. The polymer types will be aligned in each layer so that one of them can be dissolved away using its solvent to obtain square-shaped trenches with nanowires held in place by the other polymer. These trenches can be filled with Si by sputtering. The nanowires may be coated with a thin layer of insulator prior to Si sputtering to prevent carrier recombination [24] and reduce reflection [25]. Then the other polymer can be removed and the resulting trenches can be filled similarly to obtain a Si solar cell with multiple layers of embedded plasmonic nanowires. The nanowires in different layers can be fabricated at different angles with respect to each other so that randomly polarized sunlight from TM to transverse electric (TE) polarization can cause plasmon resonance in different layers. Detailed theoretical understanding of benefits of these nanowires would spur development of appropriate fabrication technology.

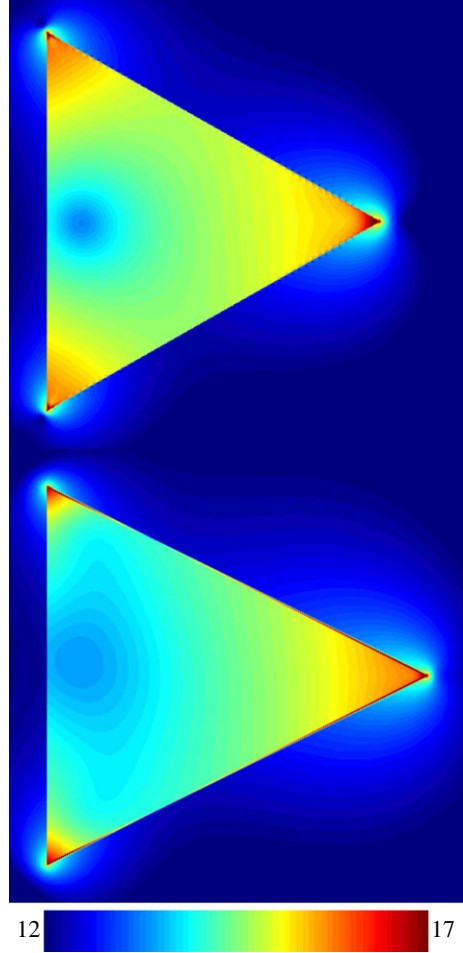

**Figure 3.** Spatial absorption maps of the nanowires at plasmon resonances in log scale. The upper figure shows the map of the equilateral nanowire at 863 nm. The lower figure shows the sharp-tipped bilateral nanowire at 840 nm.

## 3. Results and discussion

The solid lines in figure 2 show enhanced absorption in Si caused by plasmon resonance of the nanowire inclusions. The line (a1) shows absorption due to a equilateral nanowire. Its three sides were 20 nm long and the height was 17.5 nm. The line (b1) corresponds to a sharper tipped bilateral nanowire formed by increasing the height to 20 nm for the same 20 nm base. The dotted lines (a2) and (b2) represent scaled-down absorption loss in the metal of the nanowires. Scaling was done to improve the view of the graph, as absorption in metal was much larger. The line (c) shows absorption in bare Si of the simulation region without any plasmonic enhancement effect. This was obtained by excluding the nanowire from a simulation run. Comparing the absorption values in Si at the plasmon resonances of a nanowire with the absorption in bare Si at the same wavelength, absorption enhancement effect of the nanowire was estimated. Fortuitously, the nanowires showed multiple strong plasmon resonances in the wavelength range where sunlight absorbs poorly in Si. A large 368-fold absorption enhancement effect was observed at the 863 nm plasmon resonance of the equilateral nanowire. This enhancement value was much larger than the reported values of other nanostructures. For example, only 5-fold absorption enhancement was reported at 830 nm for a spherical nanoparticle embedded in Si [6]. Up to 15-fold near-bandgap enhancement was reported for core-shell type nanowires [10]. We also observed 17-fold enhancement at 768 nm for a core-shell type nanowire with a $SiO_2$ core of 5 nm radius and 6 nm thick Ag shell. Apparently, the plasmon resonance phenomena in triangular nanowires is much stronger and its shape is more effective in enhancing absorption which is clearly visible in the prominent high absorption zones in Si near the corners of the nanowires in figure 3, which shows the spatial absorption maps of the nanowires at plasmon resonances. The upper part in figure 3 shows the equilateral nanowire at 863 nm and the lower part shows the sharp-tipped bilateral nanowire at 840 nm plasmon resonance, respectively.

**Table 1.** Absorption enhancement/loss ratios. (*a*) wavelength (nm), (*b*) ratio of enhanced absorption in Si to absorption in bare Si (*c*) ratio of asorption loss in metal of nanowire to enhanced absorption in Si.

| equilateral | | | bilateral | | |
|---|---|---|---|---|---|
| (*a*) | (*b*) | (*c*) | (*a*) | (*b*) | (*c*) |
| 694 | 7.52 | 4.96 | 698 | 11.54 | 3.70 |
| 774 | 35.34 | 4.37 | 782 | 52.63 | 3.02 |
| 863 | 368.86 | 3.55 | 840 | 228.17 | 1.91 |
| 600–950 | 14.10 | 3.94 | 600–950 | 22.72 | 2.61 |

Undesirable absorption loss in the metal of the nanowire was 3.55 times larger than the absorption in Si at 863 nm. This ratio was slightly better than the ratio reported in [6]. Interestingly, it was possible to improve this ratio significantly by increasing the height of the triangle to form a sharper tip. The column (*c*) in table 1 shows lower values of absorption loss in metal to enhanced absorption in Si ratios for this nanowire. For example, absorption loss in this nanowire was 1.91 times the absorption in Si at the 840 nm plasmon resonance. This drop could be attributed to weaker plasmon resonance effect which was apparent from reduced optical absorption, specially near the corners on the left in the lower figure of figure 3. Still, strong absorption patterns can be observed near the tip on the right of this figure, which indicates that the sharper tip was capable of creating strong plasmonic field intensities even for the weaker plasmon resonance. Additionally, it can be observed that this sharper and longer tip caused enhanced absorption over a larger area in Si. The total absorption enhancement in Si due to this nanowire in the 600–950 nm wavelength range was 22.7 times compared with 14.1 times of the equilateral nanowire. These values were computed by dividing the areas under the curves (a2) and (a1) by the area under the curve (c) in figure 2. Total absorption loss in metal to enhanced absorption in Si ratio of the sharp-tipped nanowire was 2.61 compared with 3.94 of the equilateral nanowire. PC1D [26] simulations showed 100% increase of short-circuit current for the 600–900 nm spectral range of AM1.5g radiation in the typical 1.4 μm thickness of a crystal on glass type thin-film Si solar cell [27]. For this simulation, the absorption coefficients in the material absorption file was scaled by the plasmonic absorption enhancement factors at the respective wavelengths assuming that multiple layers of embedded nanowires would enhance the absorption coefficients in this way. These results clearly show that absorption enhancement and loss effects have strong dependence on nanostructure shape. We speculate that a shape-optimized three-dimensional nanoparticle with multiple sharp tips can cause larger absorption enhancement than loss even in the wavelengths where Si absorbs poorly. Moreover, if the works towards developing low-loss plasmonic materials [10] makes some progress, just the use of those materials for fabricating the nanowires discussed in this work may lead to favourable absorption enhancement to loss ratio. Large enhancement capability of these nanowires may also lead to improved performance of impurity photovoltaic devices [15,16].

# 4. Conclusion

Absorption enhancement performance of triangular Ag nanowires was evaluated for thin-film Si solar cell applications. These nanowires showed much larger absorption enhancement effects compared with other nanostructures. Additionally, absorption loss could be lowered by tuning the shape of the nanowire to form a sharper tip. This decrease was apparently due to weaker plasmon resonance effect in this nanowire. However, the sharper and longer tip was effective in creating strong plasmonic field intensities at and around the tip which caused large absorption enhancement in Si despite the weaker plasmon resonance. These results show that selection of nanostucture shape and its optimization can be important factors in achieving favourable plasmonic absorption enhancement effects in solar cell devices.

Data accessibility. Two representative datasets of spatial optical absorption data at plasmon resonances as shown in figure 3 are available from [28].

Authors' contributions. M.S.S. planned the simulations. Both authors performed the simulations, analysed the simulation results and prepared the publication.

Competing interests. We declare we have no competing interests.

Funding. We received no funding for this study.

Acknowledgements. M.S.S. thanks Prof. Dr Md. Shafiqul Islam for helpful discussions.

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
