## [Reviewer comments · Royal Society Open Science]

Review History

RSOS-191926.R0 (Original submission)

Review form: Reviewer 1

Is the manuscript scientifically sound in its present form?

Yes

Are the interpretations and conclusions justified by the results?

Yes

Is the language acceptable?

Yes

Do you have any ethical concerns with this paper?

No

Have you any concerns about statistical analyses in this paper?

No

Recommendation?

Accept with minor revision (please list in comments)

Comments to the Author(s)

This manuscript reports plasmonic absorption enhancement of triangular silver nanowires in silicon calculated by 2D FDTD simulations. The absorption enhancement of triangular silver nanowires is much larger than solid spherical nanoparticles and core-shell nanowires. Triangular nanowire with equal edge size of 20 nm showed 368 fold absorption enhancement and 3.55 times absorption loss at 863 nm. Increasing the sharpness of the corner resulted in reduced absorption loss. This work provides useful information on the shape-related plasmonic absorption enhancement and absorption loss, and will contribute to the design of photovoltaic devices. It can be published in Royal Society Open Science after marking the sizes of the objects in Figure 1.

Review form: Reviewer 2

Is the manuscript scientifically sound in its present form?

Yes

Are the interpretations and conclusions justified by the results?

Yes

Is the language acceptable?

Yes

Do you have any ethical concerns with this paper?

No

Have you any concerns about statistical analyses in this paper?

No

Recommendation?

Accept with minor revision (please list in comments)

Comments to the Author(s)

The authors reported on the absorption enhancement and loss effects of a nanostructure composed of triangular silver nanowires embedded in silicon thin film. The calculations were finished by two dimensional FDTD simulations. Two kinds of triangular silver nanowires were employed, equilateral nanowire and bilateral nanowire, the heights of which are 17.5 nm and 20 nm, respectively. The two nanowires both show large absorption enhancement effects, while the bilateral shows lower absorption loss effect. The results may be published after minor revision according to the following points.

1. Some ambiguous descriptions are shown in Abstract which should be improved. Like, 'For a nanowire with 20 nm base and 20 nm height, absorption loss was merely 1.91□ larger than the 228□ absorption enhancement at 840 nm.' in Page 2.
2. The authors mentioned the absorption is related with the electric field intensity and the imaginary component of the refractive index of the material in Line 42 Page 2. References should be added here.
3. For Simulation Methodology, the excitation mode of TFST source should be added in the Fig. 1.
4. There is a redundant (a) in the lower Fig. 1, please delete it.
5. Fig. 3 is called 'Log scale spatial absorption maps of the nanowires', while the absorption is corresponded with wavelength, which is not show in this figure. And generally, the 'Log scale' is related with electric field enhancement instead of absorption. Can the authors explain your definition in Fig. 3 more clearly and deeply?
6. It is better to show a line plot of the enhancement in Fig. 3 in order to compare the different enhancement performances more easily.

Decision letter (RSOS-191926.R0)

13-Mar-2020

Dear Dr Sabuktgin:

Title: Large Plasmonic Absorption Enhancement Effect of Triangular Silver Nanowires in Silicon
Manuscript ID: RSOS-191926

Thank you for submitting the above manuscript to Royal Society Open Science. On behalf of the Editors and the Royal Society of Chemistry, I am pleased to inform you that your manuscript will be accepted for publication in Royal Society Open Science subject to minor revision in accordance with the referee suggestions. Please find the reviewers' comments at the end of this email.

The reviewers and handling editors have recommended publication, but also suggest some minor revisions to your manuscript. Therefore, I invite you to respond to the comments and revise your manuscript.

Please also include the following statements alongside the other end statements. As we cannot publish your manuscript without these end statements included, if you feel that a given heading is not relevant to your paper, please nevertheless include the heading and explicitly state that it is not relevant to your work. We have included a screenshot example of the end statements for reference.

- Acknowledgements

- Funding statement

Please include a funding section after your main text which lists the source of funding for each author.

Because the schedule for publication is very tight, it is a condition of publication that you submit the revised version of your manuscript before 22-Mar-2020. Please note that the revision deadline will expire at 00.00am on this date. If you do not think you will be able to meet this date please let me know immediately.

- 1) A text file of the manuscript (tex, txt, rtf, docx or doc), references, tables (including captions) and figure captions. Do not upload a PDF as your "Main Document".

- 2) A separate electronic file of each figure (EPS or print-quality PDF preferred (either format should be produced directly from original creation package), or original software format)
- 3) Included a 100 word media summary of your paper when requested at submission. Please ensure you have entered correct contact details (email, institution and telephone) in your user account
- 4) Included the raw data to support the claims made in your paper. You can either include your data as electronic supplementary material or upload to a repository and include the relevant doi within your manuscript
- 5) All supplementary materials accompanying an accepted article will be treated as in their final form. Note that the Royal Society will neither edit nor typeset supplementary material and it will be hosted as provided. Please ensure that the supplementary material includes the paper details where possible (authors, article title, journal name).

Best wishes,
Dr Laura Smith
Publishing Editor, Journals

On behalf of the Subject Editor Professor Anthony Stace and the Associate Editor Professor Tobias Hertel.

RSC Associate Editor:
Comments to the Author:
(There are no comments.)

RSC Subject Editor:
Comments to the Author:
(There are no comments.)

Reviewer comments to Author:
Reviewer: 1

Comments to the Author(s)
This manuscript reports plasmonic absorption enhancement of triangular silver nanowires in silicon calculated by 2D FDTD simulations. The absorption enhancement of triangular silver

nanowires is much larger than solid spherical nanoparticles and core-shell nanowires. Triangular nanowire with equal edge size of 20 nm showed 368 fold absorption enhancement and 3.55 times absorption loss at 863 nm. Increasing the sharpness of the corner resulted in reduced absorption loss. This work provides useful information on the shape-related plasmonic absorption enhancement and absorption loss, and will contribute to the design of photovoltaic devices. It can be published in Royal Society Open Science after marking the sizes of the objects in Figure 1.

Reviewer: 2

Comments to the Author(s)

The authors reported on the absorption enhancement and loss effects of a nanostructure composed of triangular silver nanowires embedded in silicon thin film. The calculations were finished by two dimensional FDTD simulations. Two kinds of triangular silver nanowires were employed, equilateral nanowire and bilateral nanowire, the heights of which are 17.5 nm and 20 nm, respectively. The two nanowires both show large absorption enhancement effects, while the bilateral shows lower absorption loss effect. The results may be published after minor revision according to the following points.

1. Some ambiguous descriptions are shown in Abstract which should be improved. Like, 'For a nanowire with 20 nm base and 20 nm height, absorption loss was merely 1.91□ larger than the 228□ absorption enhancement at 840 nm.' in Page 2.
2. The authors mentioned the absorption is related with the electric field intensity and the imaginary component of the refractive index of the material in Line 42 Page 2. References should be added here.
3. For Simulation Methodology, the excitation mode of TFST source should be added in the Fig. 1.
4. There is a redundant (a) in the lower Fig. 1, please delete it.
5. Fig. 3 is called 'Log scale spatial absorption maps of the nanowires', while the absorption is corresponded with wavelength, which is not show in this figure. And generally, the 'Log scale' is related with electric field enhancement instead of absorption. Can the authors explain your definition in Fig. 3 more clearly and deeply?
6. It is better to show a line plot of the enhancement in Fig. 3 in order to compare the different enhancement performances more easily.

Author's Response to Decision Letter for (RSOS-191926.R0)

See Appendix A.

RSOS-191926.R1 (Revision)

Review form: Reviewer 2

Is the manuscript scientifically sound in its present form?

Yes

Are the interpretations and conclusions justified by the results?

Yes

Is the language acceptable?

Yes

Do you have any ethical concerns with this paper?

No

Have you any concerns about statistical analyses in this paper?

No

Recommendation?

Accept as is

Comments to the Author(s)

The authors addressed all of my questions.

Decision letter (RSOS-191926.R1)

Dear Dr Sabuktgin:

Title: Large Plasmonic Absorption Enhancement Effect of Triangular Silver Nanowires in Silicon
Manuscript ID: RSOS-191926.R1

It is a pleasure to accept your manuscript in its current form for publication in Royal Society Open Science. The chemistry content of Royal Society Open Science is published in collaboration with the Royal Society of Chemistry. I am sorry it has taken longer than usual to be able to send you this decision.

On behalf of the Subject Editor Professor Anthony Stace and the Associate Editor Professor Tobias Hertel.

RSC Associate Editor:
Comments to the Author:

Dear Dr. Sabuktgin.

We sincerely apologise for any delays with the review and processing of your manuscript. However, we are happy to report that your manuscript can now be published as is.

With best regards,
Tobias Hertel
Associate Editor, RSOS

RSC Subject Editor:
Comments to the Author:
(There are no comments.)

Reviewer(s)' Comments to Author:
Reviewer: 2

Comments to the Author(s)
The authors addressed all of my questions.

Appendix A

Reviewer comments to Author:

Reviewer: 1

Comments to the Author(s)

This manuscript reports plasmonic absorption enhancement of triangular silver nanowires in silicon calculated by 2D FDTD simulations. The absorption enhancement of triangular silver nanowires is much larger than solid spherical nanoparticles and core-shell nanowires. Triangular nanowire with equal edge size of 20 nm showed 368 fold absorption enhancement and 3.55 times absorption loss at 863 nm. Increasing the sharpness of the corner resulted in reduced absorption loss. This work provides useful information on the shape-related plasmonic absorption enhancement and absorption loss, and will contribute to the design of photovoltaic devices. It can be published in Royal Society Open Science after marking the sizes of the objects in Figure 1.

Reply:

The sizes of the objects in Figure 1 has been marked.

Reviewer: 2

Comments to the Author(s)

The authors reported on the absorption enhancement and loss effects of a nanostructure composed of triangular silver nanowires embedded in silicon thin film. The calculations were finished by two dimensional FDTD simulations. Two kinds of triangular silver nanowires were employed, equilateral nanowire and bilateral nanowire, the heights of which are 17.5 nm and 20 nm, respectively. The two nanowires both show large absorption enhancement effects, while the bilateral shows lower absorption loss effect. The results may be published after minor revision according to the following points.

1. Some ambiguous descriptions are shown in Abstract which should be improved. Like, 'For a nanowire with 20 nm base and 20 nm height, absorption loss was merely 1.91 times larger than the 228 times absorption enhancement at 840 nm.' in Page 2.

Reply:

This sentence has been modified to 'For a nanowire with 20 nm base and 20 nm height, absorption loss was merely 1.91 times larger than the absorption in Si at the 840 nm plasmon resonance.'

2. The authors mentioned the absorption is related with the electric field intensity and the imaginary component of the refractive index of the material in Line 42 Page 2. References should be added here.

Reply:

References and formulas have been added. The word 'refractive index' was changed to 'permittivity'.

3. For Simulation Methodology, the excitation mode of TFST source should be added in the Fig. 1.

Reply:

The TFST source has been added in Fig. 1..

4. There is a redundant (a) in the lower Fig. 1, please delete it.

Reply:

It has been deleted.

5. Fig. 3 is called 'Log scale spatial absorption maps of the nanowires', while the absorption is corresponded with wavelength, which is not show in this figure. And generally, the 'Log scale' is related with electric field enhancement instead of absorption. Can the authors explain your definition in Fig. 3 more clearly and deeply?

Reply:

The caption has been changed to "Spatial absorption maps of the nanowires at plasmon resonances in log scale." The figures were plotted after taking log of absorption values at every pixel of the figure. The plasmon resonance wavelengths are mentioned in the caption.

6. It is better to show a line plot of the enhancement in Fig. 3 in order to compare the different enhancement performances more easily.

Reply:

The line plots in Fig. 2 shows the different enhancement performances.

Additionally the following three modifications were made:

1.The sentence "Absorption loss to enhancement ratios of the nanowires in Table 1 shows lower values for this nanowire." was changed to "The column (c) in Table 1 shows lower values of absorption loss in metal to enhanced absorption in Si ratios for this nanowire."

2. The words "weaker absorption" was replaced with "reduced optical absorption".

3. The words "Total absorption loss to absorption enhancement ratio of the sharp tipped nanowire" was replaced with "Total absorption loss in metal to enhanced absorption in Si ratio of the sharp tipped nanowire"

And the following sentence and reference was added:

"Dramatic enhancement of electric field intensities were reported for triangular nanowires in Ref 9."